# The American Friendship Project: A report on the status and health of friendship in America

**Natalie Pennington[1], Jeffrey A. Hall[2]\*, Amanda J. Holmstrom[3]**

**1** Department of Communication Studies, College of Liberal Arts, Colorado State University, Fort Collins, CO, United States of America, **2** Department of Communication Studies, College of Letters, Arts, and Sciences, University of Kansas, Lawrence, KS, United States of America, **3** Department of Communication, College of Communication Arts and Sciences, Michigan State University, East Lansing, MI, United States of America

\* hallj@ku.edu

**Data Availability Statement:** The data is hosted on OSF, where we have created a project with access to the data from 2022 and 2023: https://osf.io/

## Abstract

Friendship is critical for individuals' well-being, but recent efforts to characterize Americans' friendship have suggested that these relationships are in peril. The present study is a report on the methods and results of three surveys from the American Friendship Project (AFP). The goal of the AFP is to be the most accurate and most complete account of American friendship as well as its health and change over time. The AFP reports on five critical facets of social health as it relates to friendship: 1) the structural factors of friendship (e.g., who are they, how many); 2) friendship quality (e.g., satisfaction, closeness); 3) social support from friends; 4) the quantity of online and offline communication; and 5) well-being (e.g., life satisfaction, loneliness, connection). Data was collected from two national samples of American adults in 2022 and 2023 and from a large sample of college students across three universities in 2022. The key finding from this investigation is, compared to the discouraging results of other recent surveys, Americans reported having more friends and fewer were friendless. AFP results also suggest that face-to-face (FtF) meetings among friends are quite common, as are telephone calls and text messaging. College student and adult samples reported very similar attitudes and experiences with friendship overall, but students were more likely to meet friends at school and to keep them for a shorter length of time. Another key finding is Americans long for greater closeness with friends; though over 75% were satisfied with the number of friends they had, over 40% felt they were not as close to their friends as they would like. Overall, the AFP is a rich source of data that can be used to answer a multitude of questions about friendship and its connection to well-being.

## Introduction

"Of all the means to ensure happiness throughout the whole life, by far the most important is the acquisition of friends"–*Epicurus, 3rd century BCE*

Ancient cultures celebrated the virtue of friendship, and contemporary research has confirmed their prescience. The bonds of friendship, in number and quality, robustly predict happiness and life satisfaction both cross-sectionally [1–5] and over time [6]. Spending time with

2p6qh/?view_only=444f6dc38bbc49e69c9e1a89cb617f1c.

**Funding:** This paper was supported by the University of Kansas GRF #2177080 (2022); the University of Nevada, Las Vegas Faculty Opportunity Award (2022-2023) received by Natalie Pennington; and the Michigan State University ComArtSci Research and Creative Incubator and Accelerator (CRCIA) award (2022-2025) received by Amanda J. Holmstrom. No sponsors or funders played any role in the study design, data collection and analysis, decision to publish, or preparation of the manuscript.

**Competing interests:** No authors have competing interests.

friends explains variance in happiness and well-being beyond what is explained by visits with family or neighbors [3, 4]. For older adults, opportunities to spend time with friends matters more than similar amounts of social contact with neighbors, adult children, and other family [4]. Friends are crucial sources of social support and mitigate loneliness at all stages of life [6]. High quality friendship can even compensate for lower quality family and romantic relationships [7]. Showing support and concern, spending time together, and quality conversation are definitional qualities of friendship [8], and when these qualities are communicated to friends on any given day, they promote daily health and well-being [9].

Recent reports suggest that American friendship may be in peril, and that adults in the United States–both young and old–are increasingly lonely. In February 2020, a report from the National Academies of Sciences reported that 33% of Americans over 45 feel lonely, 25% of those over 65 are socially isolated, and loneliness only increased further during the pandemic [10]. Across several countries, the time spent being social is declining [11], and Americans are increasingly spending time alone [12]. The degree to which increases in loneliness and decreases in social time correspond with a decline in number and quality of friends is a matter of significant debate. Recent studies have suggested a reduction in the number of discussion partners (i.e., "with whom do you discuss important matters?") [13] and between 8% [14] and 12% of Americans have no friends [15].

Yet, examining trends from the general social survey from 1970 to 2010, Fischer [16] concluded that Americans "were no more and no less likely to be friendless" and "had roughly the same median number of friends," although he concedes that "recent cohorts may not have been as likely to have as many friends" (p. 60). Similarly, a recent report from Ajrouch et al. [17] pushed back on the narrative that friendship network size declines with age. A reduction in the number of friends and discussion partners has been challenged [18, 19], and may be a product of methodological artifacts [20], wherein how the data is collected (and questions framed) influenced estimates.

Indeed, what constitutes "friend" can vary greatly across research, including limitations on who counts as a friend (e.g., family and romantic partners are often excluded) [1], and many studies count only same-sex friends [21]. Further still, asking people to differentiate "friend" from "close friend" and "best friend" (including work framing them as strong versus weak ties) creates more inconsistencies within the literature, highlighting the inherent fuzziness that surrounds friendship [22], and the need to better situate these relations today.

These debates regarding a decline in friendship underscore the degree of public interest and concern about the state and health of American friendship today. However, to date, there are no comprehensive annual efforts to characterize the nature of American friendship and its impact on well-being. Accurate, longitudinal methods that use similar measures and procedures can establish trends and present a clear-eyed view of the state of Americans' social health compared to one-off and/or poorly conceptualized measurement.

To truly understand the degree to which characteristics of friendship affect well-being and loneliness, high quality data is needed. The American Friendship Project (AFP) endeavors to be such a source of information. The goal of the present manuscript is to introduce the AFP and report on the first, baseline two years' findings (2022–2023). We begin by proposing the aims of the AFP, then describe the methodology of the survey and key differences in the survey composition both in terms of sample (college versus adult public sample) and year (2022 versus 2023). The results of these surveys are then contrasted with past findings to consider the current health of American friendship and the key questions that should be asked regarding social connection today.

### American Friendship Project research questions

Building on research about friendship from across several disciplines, including Communication Studies, Psychology, Sociology, and Human Development and Family Studies, the goal of the American Friendship Project (AFP) is to be an accurate and complete account of American friendship, as well as its health and change over time. Compared to the study of romantic relationships and family, friendship has received less scholarly attention [2]. For example, Fischer's [16] review of national survey data, which tracked friendship characteristics over time, suggests that rich and complex data sources of friendship are rare. Recent, high-quality sources of data on friendship (e.g., UCNets 2015–2018) have been discontinued, leaving a gap in available sources of data. These data are needed to continue to unpack the nature of friendship and how it impacts well-being.

The present investigation presents the development and results of the AFP 2022 and 2023 baseline surveys, which included questions pertaining to: 1) the structural factors of friendship (e.g., who are they, how many); 2) friendship quality (e.g., satisfaction, closeness); 3) social support from friends; 4) the quantity of online and offline communication; and 5) well-being (e.g., life satisfaction, loneliness, connection). Contemporary research on social health [23] recommends that the structure, function (e.g., social support, communication), and quality of social connections each be examined to present a full account of how and why social relationships are beneficial for well-being. Each of these aspects of social connections have been associated with well-being, but they are not highly correlated with each other.

The AFP answers basic questions, such as how many friends people have, and explores how the way researchers define a "friend" may influence results. The AFP also uses established measures of social health (e.g., companionship, loneliness) and well-being (e.g., life satisfaction) to link friendship processes to well-being. In addition to these year-to-year standard questions, each year the AFP includes open-ended questions on different topics. The present manuscript describes the procedures, measures, and results of the first two years of the AFP survey, addressing at its core four primary research questions tied to our focus on relationships between friendship and social health:

**RQ1:** What is the structure of Americans' friendships (e.g., number, characteristics, initiation, longevity, closeness)?

**RQ2:** What are the communicative functions of Americans' friendships (e.g., frequency, modality, proximity)?

**RQ3:** How do Americans perceive the quality of their friendships (e.g., support, satisfaction, time, closeness)?

**RQ4**: What is the state of Americans social health and well-being (e.g., connection, companionship, loneliness, disconnection)?

## Materials and methods

In the summer of 2022, the three co-project leaders commissioned a quota sample of Americans from Siena College Research Institute (SCRI), a national leader in public opinion polling. SCRI contracts with the national survey panel company Lucid, who has established a rigorous process for data quality checks [24] and has been identified as a useful source of data collection to reach a demographically diverse population in the United States [25]. For the first year of data collection, IRB approval was obtained for the project by the University of Kansas (IRB #STUDY00147041). In the first year of the AFP, data collection through SCRI was conducted

between July 21 and August 1, 2022. Subsequent years of public AFP data had additional IRB approval from the University of Nevada, Las Vegas (IRB #UNLV-2022-423) and Michigan State University (IRB #STUDY00008066). In the second year of data collection, data were collected between June 9–24, 2023. In the fall of 2022, the three co-project leaders fielded the survey at their respective home universities located in the Midwest, the upper Midwest, and the West. Individual IRB approval was obtained for students from each university (Kansas IRB #STUDY00147041, Michigan State IRB #STUDY00008066, and the UNLV, IRB#UNLV-2022-396). College student data collection was conducted between September 16 and December 2, 2022. The purposes of collecting data for the college sample were as follows: a) although loneliness is often high among young adults, there is recent evidence that loneliness is even higher among recent generations of young adults [26]; b) emerging adulthood is a crucial period of social development due to changing contexts and maturation [27]; and c) negative outcomes are often linked to computer-mediated communication preferences among young adults [11]. In addition, the 2022 student sample was invited to participate in a longitudinal component of the AFP to track changes over time.

All participants were first provided with a digital consent statement that they were asked to click yes or no for. Participants who declined were directed out of the study and thanked for their time. If they consented, they were directed to a series of demographic measures, then completed a name generator task, followed by questions about friendship generally, and finally answered questions about social and global well-being. These measures are explicated below and detailed for comparison of 2022 and 2023 in Table 1.

## Measures

**Survey section I: Demographics.** At the start of the survey, participants answered several demographic questions which included: age, gender, sexual orientation, ethnicity, race, education level, relational status, living arrangements (alone or with others and whom), and employment status. Prior to nominating friends, participants were also asked whether they had experienced any of a possible twelve life changes in the past year (2022 survey). Items were generated from the existing literature and were included based on evidence that life changes often spur changes in one's social network [28]. The item "started a new job/school" was split into two categories in 2023, yielding 13 options. See Table 2 for the full list of life changes assessed, as well as an additional option to select "no changes".

**Survey section II: Friend name generation task.** In Year 1, we chose to let participants define friendship by asking them to complete an open-ended question to begin the survey where they were asked to think about what it means to be a friend and to share three characteristics that would lead them to call someone a friend. Subsequently, respondents were given a name generator task. They were asked to fill in the blank with the first name or initials of up to seven (7) people that they would say are their friends. If they indicated they did not have at least one friend, they were prompted, "You indicated that you can't name anyone who fits that definition. Perhaps there are people in your life with whom you have regular pleasant or enjoyable interactions when you see them." If they indicated that there was no one who fit that definition, they were asked to confirm that they had no one who met either definition, and were given two options: to complete another part of the survey that would take about the same amount of time as the survey on friendship or to return to an earlier section because they can remember some people who they would call a friend. Using several prompts to improve participant recall and reduce non- response is recommended practice when using name generator tasks and allowed us to gather data from those who said they had no friends but did have people with whom they had regular, enjoyable interactions [19, 20].

**Table 1.  AFP survey items and flow.**

| Survey Section | 2022 Measures | 2023 Measures |
|---|---|---|
| **Section I:** Demographics | Age*<br>Gender*<br>Sexual Orientation*<br>Ethnicity*<br>Race*<br>Education Level**<br>Income**<br>Relational Status*<br>Living Arrangements*<br>Employment Status*<br>Life Changes* | Age*<br>Gender*<br>Sexual Orientation*<br>Ethnicity*<br>Race*<br>Education Level**<br>Income**<br>Relational Status*<br>Living Arrangements*<br>Employment Status*<br>Life Changes* |
| **Section II:** Friend Name Generation Task | What is a friend? (open text)<br>Number of friends up to 7*<br>Names/initials of those up to 7 friends | How many friends? (up to 51+)*<br>Names/initials of up to 5 friends*<br>Last year: new friend?<br>Last year: lost touch with a friend? |
| **Section IIIA:** Individual Friendships<br>This section was completed by anyone who said they had friends or had people they had regular enjoyable interactions with. All items were asked for each friendship. | Friend Demographics*<br>Relationship Type*<br>How they Met*<br>Friendship Length*<br>Emotional Closeness<br>Closeness: change last year?<br>Ambivalence<br>FTF Frequency*<br>FTF: change last year?<br>Media Frequency*<br>Most supportive thing last year? (open text) | Friend Demographics*<br>Relationship Type*<br>How they Met*<br>Friendship Length*<br>Emotional Closeness<br>Closeness: change last year?<br>Ambivalence<br>FTF Frequency*<br>FTF: change last year?<br>Media Frequency*<br>*New Friend asked the same items except for change in the last year and friendship length.*<br>*Lost Friend asked demographics, relationship type, how met, friendship length, reason for loss, who ended relationship, loss completeness, satisfaction with loss, entropy.* |
| *OR* | | |
| **Section IIIB:** No Friendships<br>This section was completed by participants who indicated they did not have a friend. | Why do you think you have no friends? (open text)<br>Have you ever had a friend? (yes/no)<br>If yes: Friend demographics & why did it end?<br>Maximizing Tendency in Friendship Selection<br>Dispositional Preference for Solitude<br>Friendship Assessment<br>Purpose in Life<br>Global Attachment | Why do you think you have no friends? (open text)<br>Have you ever had a friend? (yes/no)<br>If yes: Friend demographics & why did it end?<br>Maximizing Tendency in Friendship Selection<br>Dispositional Preference for Solitude<br>Friendship Assessment<br>Purpose in Life<br>Entropy<br>Actual Time with Different Network Members<br>Ideal Time with Different Network Members<br>Experience deciding not to be friends? (open text) |
| **Section IV:** General Friendship<br>This section was completed by anyone who said they had friends or had people they had regular enjoyable interactions with. | Density of Contacts*<br>Overall Satisfaction with Friendships*<br>Ease Making and Maintaining Friendships*<br>Social Support*<br>COVID-19 Impact on Friendship*<br>Last Time They Made a New Friend*<br>Lost Touch with a friend in the last year (yes/no) *<br>If yes, multiple friends or one? (yes/no) * | Density of Contacts*<br>Overall Satisfaction with Friendships*<br>Ease Making and Maintaining Friendships*<br>Social Support*<br>Actual Time with Different Network Members<br>Ideal Time with Different Network Members<br>Experience deciding not to be friends? (open text)<br>Social Support Gaps (scale)<br>Social Support Gaps (open text) |

*(Continued)*

**Table 1.** (Continued)

| Survey Section | 2022 Measures | 2023 Measures |
|---|---|---|
| **Section V:** Well-Being | Life Satisfaction*<br>Loneliness*<br>State Social Connection*<br>State Social Disconnection*<br>Perceived Stress*<br>Companionship*<br>Perceived Support from Friends* | Life Satisfaction*<br>Loneliness*<br>State Social Connection*<br>State Social Disconnection*<br>Perceived Stress*<br>Companionship*<br>Perceived Support from Friends* |

**Notes.** *Items marked with a single asterisk are detailed further within the current manuscript. **Items with a double asterisk are included in supplemental online materials.

In Year 2, we excluded the open-ended question about friendship definitions to make room for additional open-ended questions about social support (to be described later). In addition, to increase an understanding of the upward bounds of friendship, in 2023 we included a single item that asked participants to identify the total number of friends they believed they had, on the following ordinal scale: 0 = no friends, 1 = 1–5 friends, 2 = 6–10 friends, 3 = 11–15 friends, 4 = 16–20, 5 = 21–30, 6 = 31–50 friends, 7 = 51+ friends. This item sought to capture the upper limits of friendship network size, which was limited to only seven in 2022. However, like the first year, if they selected no friends, they were asked if they had people with whom they had regular enjoyable interactions with the same prompts and survey flow.

Another significant change between 2022 and 2023 was the number of friends possible in the name generator was capped at seven in 2022 and at five in 2023. The maximum number was reduced between years for two reasons. First, it allowed for the opportunity to go more in depth about different types of friendship experiences. In 2023, participants also were asked if they had made a new friend in the last year that they had not already listed in their first five names, as well as experiences with friendships that ended. In this regard, participants still shared in depth on up to seven friendships, but two of them were specifically targeted at a new friendship and an ended friendship. The second reason was that the introduction of the higher cap question (between 0 and 51+ friends) allowed for greater clarity on the overall size of one's network, versus a cap of seven from year one. In 2023, many participants ($n = 640$) indicated they had made a new friend, and this number was added to the total number of friends an individual had. Similarly, many participants ($N = 804$) had lost touch with a friend in the last year. This was not added to the total number of friends, leaving the total possible for 2023 at six.

**Survey section III: Individual friendships, no friendships, and ended friendships.** For each name generated, including new friends, we asked participants to identify that friend's relationship to the participant, gender identity, sexual orientation, ethnicity, race, age, where they met, and where they live in relation to the participant. We also asked about friendship length, feelings of ambivalence toward the friend, closeness, changes in closeness over the past year, and frequency of communication via the following channels: face-to-face (FtF), voice calls, video calls, email, texting or direct messaging, and following/engagement on social media on a 9-point scale (0 = Never, 8 = several times a day). We also asked the extent to which FtF communication in the past year was similar to previous years. In 2022, special focus was given to high-quality social support from friends, and participants were provided with all the names they had given us and asked to identify (through an open-text response) the single most supportive thing one of those friends had done for them in the last year. In 2023, our topical focus was on social support gaps (i.e., discrepancies between desired and received support); participants were asked to share a time where they had a problem in the last month

**Table 2.** AFP 2022 and 2023 study participant demographics Part III.

| Variable | 2022 Student Sample | | 2022 SCRI Sample | | 2023 SCRI Sample | |
|---|---|---|---|---|---|---|
| **Number of Friends*** | N | % | N | % | N | % |
| Zero | 5 | .4 | 78 | 3.2 | 33 | 1.5 |
| No Friends, but Enjoyable Interactions | 14 | 1.2 | 95 | 3.9 | 64 | 2.9 |
| One | 33 | 2.9 | 456 | 18.8 | 97 | 4.4 |
| Two | 42 | 3.6 | 348 | 14.4 | 142 | 6.4 |
| Three | 82 | 7.1 | 430 | 17.8 | 183 | 8.3 |
| Four | 97 | 8.4 | 293 | 12.1 | 150 | 6.8 |
| Five | 117 | 10.2 | 155 | 6.4 | 1073 | 48.6 |
| Six | 73 | 6.3 | 73 | 3.0 | 565 | 25.6 |
| Seven | 703 | 61.0 | 587 | 24.3 | -- | -- |
| **Total Number of Friends (2023)**** | | | | | | |
| No Friends | -- | -- | -- | -- | 33 | 1.5 |
| No Friends, but Enjoyable Interactions | -- | -- | -- | -- | 64 | 2.9 |
| 1–5 Friends | -- | -- | -- | -- | 995 | 44.4 |
| 6–10 Friends | -- | -- | -- | -- | 543 | 24.2 |
| 11–15 Friends | -- | -- | -- | -- | 222 | 9.9 |
| 16–20 Friends | -- | -- | -- | -- | 153 | 6.8 |
| 21–30 Friends | -- | -- | -- | -- | 79 | 3.5 |
| 31–50 Friends | -- | -- | -- | -- | 61 | 2.7 |
| 51+ Friends | -- | -- | -- | -- | 93 | 4.1 |
| **Life Changes (all that apply)*****  | | | | | | |
| Gotten Married | 3 | .3 | 90 | 3.7 | 51 | 2.3 |
| Started a New Job-School | 765 | 66.4 | 381 | 15.7 | -- | -- |
| Started a New Job | -- | -- | -- | -- | 457 | 20.4 |
| Started a New School | -- | -- | -- | -- | 100 | 4.5 |
| Stopped Working-Retired | 147 | 12.8 | 147 | 6.1 | 157 | 7.0 |
| Moved to a New Place | 513 | 44.5 | 432 | 17.9 | 458 | 20.4 |
| Adopted or Had Child(ren) | 6 | .5 | 51 | 2.1 | 54 | 2.4 |
| Child(ren) Moved Out of the Home | 5 | .4 | 49 | 2.0 | 54 | 2.4 |
| Death of a Romantic Partner | 4 | .3 | 42 | 1.7 | 37 | 1.6 |
| Death of a Household Member | 43 | 4.6 | 153 | 6.3 | 140 | 6.2 |
| Moved to a New City, State, or Country | 287 | 24.9 | 241 | 10.0 | 238 | 10.6 |
| Started a New Romantic Relationship | 270 | 23.4 | 193 | 8.0 | 199 | 8.9 |
| Ended a Romantic Relationship | 219 | 19.0 | 148 | 6.1 | 155 | 6.9 |
| Graduated from School | 293 | 25.4 | 123 | 5.1 | 136 | 6.1 |
| None of the Above | 167 | 14.5 | 1324 | 54.7 | 1118 | 49.8 |

**Notes.** 2022 Student $N$ = 1152. 2022 SCRI $N$ = 2420. 2023 SCRI $N$ = 2243. *Seven is the highest number participants could identify in 2022, Six is the highest number reported on for 2023. **In 2023 we added the question: about how many total friends do you have (ranging from 0 to 51+). ***For life changes, Started a New Job-School was a single option in 2022 and was split into two choices for 2023.

(open-text response), followed by a reflection on the potential for support gaps [29] within their network. To conclude this section in 2023, participants were shown their answers to the support received and asked to reflect, in an open-text response, about that experience.

Individuals who reported having no friends and no enjoyable interaction partners were asked an open-ended item about why they believed they did not have friends. They responded to a question about whether they had ever had a friend, and if so, what led to the end of that

Table 3. AFP 2022 and 2023 study friendship experiences.

| Items | 2022 Student Sample | | 2022 SCRI Sample | | 2023 SCRI Sample | |
|---|---|---|---|---|---|---|
| | M(SD) | % Agree | M(SD) | % Agree | M(SD) | % Agree |
| I'm not as close to my friends as I would like to be. | 3.53(1.75) | 36.8 | 3.86(1.82) | 40.1 | 4.04(1.74) | 47.3 |
| My friends and I aren't getting along. | 1.86(1.08) | 3.7 | 2.18(1.59) | 9.9 | 2.12(1.33) | 6.7 |
| My friends celebrate my good news. | 6.14(.93) | 95.4 | 5.80(1.25) | 84.0 | 5.78(1.17) | 86.7 |
| It is difficult to maintain friendships at this time in my life. | 3.64(1.85) | 40.8 | 3.39(1.93) | 32.1 | 3.46(1.89) | 36.0 |
| I am satisfied with the number of friends I have. | 5.35(1.62) | 75.4 | 5.51(1.53) | 76.4 | 5.41(1.56) | 76.2 |
| I am satisfied with the amount of time I am able to spend with my friends. | 4.25(1.83) | 51.1 | 4.74(1.79) | 58.2 | 4.58(1.81) | 56.6 |
| It is difficult to make new friends. | 4.36(1.96) | 56.2 | 4.12(1.97) | 46.6 | 4.16(1.93) | 48.9 |
| It was easier to make friends at another time in my life. | 4.76(1.97) | 62.2 | 4.72(1.77) | 58.3 | 4.96(1.76) | 65.6 |
| In the past year, maintaining close relationships has been difficult or frustrating for me. | 3.67(1.76) | 37.3 | 3.52(1.84) | 33.9 | 3.68(1.79) | 37.1 |
| In the past year, I am satisfied with how well I have maintained friendships. | 4.98(1.49) | 69.5 | 5.09(1.57) | 68.4 | 4.89(1.61) | 62.9 |
| I missed out on making friends as a result of the COVID-19 pandemic.* | 4.49(1.97) | 57.7 | 3.66(1.93) | 36.4 | -- | -- |
| The COVID-19 pandemic strengthened my friendships.* | 4.36(1.63) | 51.6 | 4.30(1.62) | 41.2 | -- | -- |
| I lost friends as a result of the COVID-19 pandemic.* | 3.80(1.98) | 42.5 | 3.00(1.91) | 24.9 | -- | -- |
| I am uncertain how to interact around my friends.** | -- | -- | -- | -- | 2.65(1.68) | 17.1 |

Notes. 2022 Student $N$ = 1138. 2022 SCRI $N$ = 2340. 2023 SCRI $N$ = 2210. Responses shared apply only to participants who indicated they have at least one friend. All items were scaled from 1 (strongly disagree) to 7 (strongly agree). *These items were only asked in 2022. **These items were only asked in 2023.

friendship (open-ended text response question). In 2023, we added questions about the lost friend's demographics, length of the lost friendship, satisfaction with current conflict and completeness of friendship dissolution, and a modified 7-item version of the Relational Entropy Scale [30]. In both years, this group of participants were also asked how important it was for them to have friends. They completed the Maximizing Tendency in Friendship Selection Scale [31], the Dispositional Preference for Solitude Scale [32], the 9-item Experiences in Close Relationships–Relationship Structures Questionnaire [33], and the Purpose in Life Scale [34]. The results of survey responses for participants indicating they had no friends are reported elsewhere.

**Survey section IV: General friendship measures.** In the next section of the survey, participants with at least one friend completed a series of general friendship measures which included: overall satisfaction with friendships, ability to share good news with friends, ease in making and maintaining friendship, and (in 2022 only) the impact of the COVID-19 pandemic on friendship. See Table 3 for the exact wording for each item, drawn in part from Friendship Network Satisfaction Scale [35]. In 2022, participants were also asked in this section about their density of contacts, when they last made a friend and if they had lost touch with a friend in the past year (shared in Table 4).

In 2023, we also sought more information about changes in friendship closeness. Many participants in 2023 indicated they had lost touch with a friend in the past year ($n$ = 804), which allowed for additional analysis. Participants were asked to identify the reasons why they felt they had lost touch, their relationship to the person before losing touch, demographic characteristics of the friend (gender, sexual orientation, ethnicity, race, age), how they had met, their proximity to the person, how long they were friends, if they had completely lost touch, and if they were satisfied with their current level of communication with the former friend. Participants completed a modified 7-item version of the Relational Entropy Scale [30]. They were also asked to reflect on how much time they currently spend with different subsets of their

**Table 4. AFP 2022 and 2023 study demographics friends Part II.**

| Variable | 2022 Student Sample | | 2022 SCRI Sample | | 2023 SCRI Sample | |
|---|---|---|---|---|---|---|
| **Type of Friendship (all that apply)** | N | % | N | % | N | % |
| Best Friend | 1931 | 28.8 | 2291 | 25.6 | 2162 | 22.4 |
| Close Friend | 2925 | 43.7 | 3046 | 34.1 | 3417 | 35.4 |
| Friend | 1448 | 21.6 | 1825 | 20.4 | 2554 | 26.5 |
| Casual Friend | 429 | 6.4 | 810 | 9.1 | 1425 | 14.8 |
| Colleague-Work Friend | 127 | 1.9 | 215 | 2.4 | 416 | 4.3 |
| Romantic Partner | 319 | 4.8 | 575 | 6.4 | 449 | 4.7 |
| Parent | 51 | .8 | 156 | 1.7 | 88 | .9 |
| Child | 4 | .1 | 331 | 3.7 | 154 | 1.6 |
| Sibling | 196 | 2.9 | 389 | 4.4 | 266 | 2.8 |
| Other | 149 | 2.2 | 298 | 3.3 | 225 | 2.3 |
| **Friendship Length[+]** | | | | | | |
| Less than 1 year | 1127 | 16.9 | 368 | 4.1 | 661 | 6.9 |
| 1–2 years | 1414 | 21.2 | 787 | 8.8 | 1156 | 12.0 |
| 3–4 years | 1396 | 20.9 | 1157 | 12.9 | 1396 | 14.5 |
| 5–10 years | 1771 | 26.6 | 1907 | 21.3 | 2124 | 22.0 |
| 11–20 years | 865 | 13.0 | 1686 | 18.9 | 1717 | 17.8 |
| 20 + years | 93 | 1.4 | 3031 | 33.9 | 2594 | 26.9 |
| **Friendship Co-Location** | | | | | | |
| Live Together | 918 | 13.7 | 863 | 9.7 | 529 | 5.1 |
| Same Neighborhood | 764 | 11.4 | 1103 | 12.3 | 1471 | 14.6 |
| Same Town/City | 2823 | 42.2 | 2740 | 30.7 | 3444 | 34.1 |
| Different Town/City (Same State) | 882 | 13.2 | 2401 | 26.9 | 2656 | 26.3 |
| Different State (Same Country) | 1095 | 16.4 | 1617 | 18.1 | 1773 | 17.6 |
| Different Country | 184 | 2.7 | 212 | 2.4 | 221 | 2.2 |
| **How Friends Met (all that apply)** | | | | | | |
| Through Family/They are Family* | 698 | 10.4 | 1902 | 21.3 | -- | -- |
| Through Family (2023) | -- | -- | -- | -- | 915 | 8.9 |
| They are Family (2023) | -- | -- | -- | -- | 769 | 7.5 |
| Through School | 4366 | 65.2 | 1816 | 20.3 | 2123 | 20.6 |
| Through Work | 456 | 6.8 | 1802 | 20.2 | 1285 | 12.5 |
| Through Place of Worship | 100 | 1.5 | 475 | 5.3 | 456 | 4.4 |
| Through Neighborhood | 321 | 4.8 | 914 | 10.2 | 1471 | 13.8 |
| Through Club-Organization | 643 | 9.6 | 399 | 4.5 | 429 | 4.2 |
| Through Sports Team** | 59 | .9 | 24 | .3 | 203 | 2.0 |
| Online | 395 | 5.9 | 510 | 5.7 | 608 | 5.9 |
| Through Another Friend | 1305 | 19.5 | 1129 | 12.6 | 1760 | 17.1 |
| Through Romantic Partner | 73 | 1.1 | 315 | 3.5 | 396 | 3.8 |
| Through Child | 4 | .1 | 161 | 1.8 | 263 | 2.6 |
| Other | 216 | 3.2 | 623 | 7.0 | 300 | 2.9 |

**Notes.** 2022 Student Friend *N* = 6667. 2022 SCRI Friend *N* = 8936. 2023 SCRI Friend *N* = 10288. *In 2023 through family and are family were split into two separate categories. **Denotes a coded category added based on open-text "other" responses within that variable in 2022, participants in this category are still included in "other" in 2022. This was added as an option in 2023. + Length of friendship not asked for new friends in 2023.

**Table 5. AFP 2022 and 2023 study friendship well-being variables.**

| Variable | Source | Scale Range | # items | α | 2022 Student Sample M(SD) | α | 2022 SCRI Sample M(SD) | α | 2023 SCRI Sample M(SD) |
|---|---|---|---|---|---|---|---|---|---|
| Life Satisfaction | Diener et al. (1985) [36] | 1–7 (Disagree-Agree) | 5 | .84 | 4.42(1.30) | .89 | 4.37(1.52) | .92 | 4.35(1.56) |
| Loneliness | Hughes et al. (2004) [37] | 1–3 (Hardly Ever-Often) | 3 | .83 | 1.92(.61) | .87 | 1.74(.68) | .86 | 1.76(.67) |
| State Connection | Lok & Dunn (2023) [38] | 1–7 (Disagree-Agree) | 6 | .86 | 5.32(.99) | .90 | 4.95(1.28) | .90 | 4.97(1.27) |
| State Disconnection | Lok & Dunn (2023) [38] | 1–7 (Disagree-Agree) | 4 | .89 | 3.91(1.53) | .92 | 3.66(1.73) | .93 | 3.75(1.74) |
| Perceived Stress | Warttig et al. (2013) [39] | 1–5 (Never-Always) | 4 | .71 | 2.80(.66) | .80 | 2.64(.86) | .80 | 2.76(.84) |
| Companionship | Cyranowski et al. (2013) [40] | 1–5 (Never-Always) | 6 | .88 | 4.08(.77) | .94 | 3.71(1.06) | .94 | 3.77(1.01) |
| Perceived Support | Zimet et al. (1998) [41] | 1–7 (Disagree-Agree) | 4 | .86 | 6.09(.82) | .90 | 5.84(1.08) | .91 | 5.68(1.14) |

**Notes.** 2022 Student $N$ = 1142. 2022 SCRI $N$ = 2111. 2023 SCRI $N$ = 2243.

social network and how they would ideally prefer to spend their time. Finally, participants reflected in response to an open-ended question about a time in which they met someone but ultimately decided not to become friends with that person, and the reason(s) why.

**Survey section V: Well-being measures.** The survey concluded with all participants answering a series of well-being measures, which included: life satisfaction [36]; loneliness [37]; state social connection and disconnection [38]; perceived stress [39]; companionship [40]; and perceived support from friends [41]. Table 5 shares the scaling and reliability information for each of these measures.

**Data cleaning.** Participants who did not complete at least 75% of the survey were removed first. Several procedures were then used to screen the quality of the remaining responses. Data was flagged if respondents failed to respond correctly to two attention check items. Data was also screened for straight-lining (i.e., identical, non-midpoint response to a positively and negatively worded

question from the same scale). Open-ended responses (e.g., friend definition, name

listing task, social support) served as additional data quality checks. The name generator responses were screened for names of celebrities or politicians. Responses were flagged if they were nonsense (e.g., used words from the question without answering it); used sequential names or numbers (e.g., good better best; 1 2 3); and/or that clearly did not answer the question. Responses of "I don't know", "unsure," "IDK," "I don't want to say", however, were retained. Responses that listed suspicious names and failed one or more of the two attention checks or that did not finish the survey were deleted ($n$ = 290). Responses that (a) listed a suspect name and (b) failed one or two attention checks and/or failed a straight-lining check were then deleted ($n$ = 81). Responses with a nonsense response to either of the two open- ended questions and another flagged response were then deleted ($n$ = 575). Responses with *only* "I don't know" responses and another flagged response were then deleted ($n$ = 31). Not including participants who did not complete at least 75% of the survey, the final retention rate was 71.2%. The final number of participants in 2022 was $N$ = 2,420.

The same initial procedures for data cleaning were also used in the second year of data collection (e.g., completion rate, straight lining). From the initial data set, responses to the survey completed in under 500 seconds ($n$ = 168), duplicate responses ($n$ = 12), and those who did not consent to participate ($n$ = 180) were removed. Data were next screened using three attention check items, including an open-text box that instructed participants to write nothing and move to the next page, as well as two questions where they were asked to pick the correct answer (e.g., "select 'daily' from the list below"). Of note, 452 responses failed the open-text

screener, and at least two additional screening checks, serving as one of the strongest predictors of data quality. Additional open-ended responses (e.g., name listing, social support) served as further data quality checks. The name generator responses were screened for names of celebrities or politicians, and nonsensical responses (e.g., "no", "Walmart", numbers). Responses failing three or more total checks were deleted. The final number of participants in 2023 was $N = 2{,}243$, with a retention rate of 68.3%.

The responses from the student surveys were screened using similar procedures. A total of 69 responses failed multiple data screening checks and were removed from the sample. The final number of participants was $N = 1{,}152$, with a retention rate of 94.3%.

**Data recoding.**   The use of "other" response options with allowance for open text produced new categories for inclusion in the 2023 survey. In 2022 a high number of participants who had selected "other" for race wrote variations of: Hispanic, Latino/x/a, Chicano/a, Mexican. A new category for race reflects this self-identified category for both years. For gender, non-binary was a new category added, and for sexual orientation the responses asexual and pansexual were added in 2023, based on open-text answers. For employment status, three additional categories were added in 2023 based on open-text responses: disabled, self-employed, and homemaker.

## Results and discussion

In the sections that follow we detail and offer brief comparisons across each of the five sections of the AFP data collected, with a primary focus on the structure of friendship (RQ1), communication with friends (RQ2), the quality of friendship (RQ3), and well-being and social health (RQ4). Each table reports across the three samples obtained: the 2022 SCRI data, 2023 SCRI data, and the 2022 college sample. The reporting below does not include the entirety of the survey items, focusing instead on key questions across each section, scales reported on both years, as well as data related to participants who indicated at least one friend. None of the open-ended responses are reported here.

### Participant demographics

To begin, we share information about our participants across each year before turning to the research questions at hand. Demographic information by sample is located in Tables 6 and 7, including participant age, gender, sexual orientation, ethnicity, race, relational status, living arrangements, and employment status. Information about income and education level are reported in Supplemental Table A in S1 File. Participant demographics for the SCRI data across both years largely match United States Census data [42], although in both cases the AFP data had a higher total of female participants (57.2–57.4% versus 50.4% for Census).

### RQ1: The structure of friendship

The vast majority of respondents indicated that they had at least one friend ($> 92.1\%$ across samples), and an additional 1.2% to 3.9% identified someone with whom they had "regular pleasant or enjoyable interactions." Of the participants who filled in names for the "regular pleasant or enjoyable interactions" prompt, they indicated that 63.2% of the names they listed were friends of some type (i.e., close, casual), which suggests the true rate of no friends is between 2 and 3%. Nearly all 'no friends' participants were from the SCRI samples, with a very small percentage of the college sample reporting no friends (0.4%).

In 2023, participants were asked how many friends they had on an ordinal scale. Similar to the name listing results, only 1.5% indicated they had no friends and no enjoyable interaction partners. The modal number was between 1–5 friends (44.4%), with 24.2% indicating they had

**Table 6. AFP 2022 and 2023 study participant demographics Part I.**

| Variable | 2022 Student Sample | | 2022 SCRI Sample | | 2023 SCRI Sample | |
|---|---|---|---|---|---|---|
| | Range | M(SD) | Range | M(SD) | Range | M(SD) |
| **Age** | 18–65 | 20.07(3.44) | 18–93 | 46.47(17.26) | 18–95 | 44.83(16.73) |
| **Gender** | N | % | N | % | N | % |
| Female | 697 | 60.5 | 1384 | 57.2 | 1288 | 57.4 |
| Male | 428 | 37.2 | 1023 | 42.3 | 935 | 41.7 |
| Nonbinary* | 13 | 1.1 | 3 | .1 | 10 | .4 |
| Transgender Male | 5 | .4 | 5 | .2 | 7 | .3 |
| Transgender Female | 1 | .1 | 1 | .0 | 1 | .0 |
| Prefer to be identified as. . . | 21 | 1.8 | 1 | .1 | 2 | .1 |
| **Sexual Orientation** | | | | | | |
| Heterosexual | 918 | 79.7 | 2163 | 89.4 | 1953 | 87.1 |
| Bisexual | 128 | 11.1 | 159 | 6.6 | 157 | 7.0 |
| Gay | 22 | 1.9 | 31 | 1.3 | 43 | 1.9 |
| Lesbian | 16 | 1.4 | 30 | 1.2 | 31 | 1.4 |
| Don't Know | 28 | 2.4 | 17 | .7 | 0 | 0 |
| Pansexual* | 13 | 1.1 | 4 | .2 | 22 | 1.0 |
| Asexual* | 12 | 1.0 | 4 | .2 | 15 | .7 |
| Prefer to be identified as. . . | 40 | 3.5 | 5 | .2 | 5 | .2 |
| **Ethnicity** | | | | | | |
| Hispanic/Latino/a/x | 197 | 17.1 | 301 | 12.4 | 242 | 10.8 |
| Not Hispanic/Latino/a/x | 916 | 79.5 | 2119 | 87.6 | 2001 | 89.2 |
| No Response | 39 | 3.4 | 0 | 0 | 0 | 0 |
| **Race (all that apply)** | | | | | | |
| White/Caucasian | 795 | 69.0 | 1826 | 75.5 | 1808 | 80.6 |
| Black/African American | 109 | 9.5 | 385 | 15.9 | 297 | 13.2 |
| Asian | 216 | 18.8 | 103 | 4.3 | 87 | 3.9 |
| Other (Hispanic)* | 38 | 3.3 | 70 | 2.9 | 26 | 1.2 |
| American Indian/Alaskan Native | 20 | 1.7 | 84 | 3.5 | 68 | 2.6 |
| Native Hawaiian/Pacific Islander | 36 | 3.1 | 15 | .6 | 14 | .6 |
| Prefer to be identified as. . . | 55 | 4.8 | 101 | 4.2 | 46 | 2.1 |

**Notes.** 2022 Student $N$ = 1152. 2022 SCRI $N$ = 2420. 2023 SCRI $N$ = 2243.

*Denotes a coded category added based on open-text "other" responses within that variable in 2022, participants in these added categories are still included in "other" in 2022. These responses were added as formal options in 2023: Nonbinary (gender), Pansexual and Asexual (Sexual Orientation).

6–10 friends. The number of friends identified by SCRI participants in 2023 in the name generator task was much higher than it was in 2022. In 2022, 33.7% listed five or more friends. In 2023, 48.6% of participants listed five friends and an additional 25.6% listed six. The number of friends listed in 2023 by the SCRI sample was similar to the 2022 student sample (77.5% with five or more friends).

Table 8 reports individual demographic characteristics of friends reported in the name generation task. Table 4 reports information about participants' friends (as reported by participants) in terms of the nature of their relationship. Participants were allowed to identify their friends with as many categories as they chose. The most common categories were close friend (34.1% to 43.7%), best friend (22.4% to 28.8%), and friend (20.4% to 26.5%). Casual friends were 14.8% of the names listed in 2023 compared to 9.1% of the 2022 SCRI sample and 6.4% of the 2022 student sample. Regardless, this suggests that at least 75% of all friends listed were

**Table 7.** AFP 2022 and 2023 study participant demographics Part II.

| Variable | 2022 Student Sample | | 2022 SCRI Sample | | 2023 SCRI Sample | |
|---|---|---|---|---|---|---|
| **Relational Status** | **N** | **%** | **N** | **%** | **N** | **%** |
| Single | 697 | 60.5 | 687 | 28.4 | 641 | 28.6 |
| Married | 23 | 2.0 | 1019 | 42.1 | 914 | 40.7 |
| Dating | 424 | 36.8 | 249 | 10.3 | 221 | 9.9 |
| Separated | 1 | .1 | 131 | 5.4 | 30 | 1.3 |
| Engaged | 4 | .3 | 55 | 2.3 | 68 | 3.0 |
| Divorced | 1 | .1 | 40 | 1.7 | 241 | 10.7 |
| Widowed | 0 | 0 | 8 | .3 | 108 | 4.8 |
| Other | 2 | .2 | 231 | 9.5 | 20 | .9 |
| **Living Arrangements (all that apply)** | | | | | | |
| Alone | 83 | 7.2 | 581 | 24.0 | 490 | 21.8 |
| Romantic Partner/Spouse | 76 | 7.1 | 1292 | 53.4 | 1209 | 53.9 |
| Extended Family | 437 | 40.9 | 509 | 27.7 | 475 | 21.2 |
| Child(ren) under 18 | 144 | 13.5 | 749 | 31.0 | 778 | 34.7 |
| Roommates | 633 | 59.2 | 207 | 8.6 | 182 | 8.1 |
| **Employment Status** | | | | | | |
| Unemployed | 560 | 48.6 | 1062 | 43.9 | 916 | 40.8 |
| Full Time | 102 | 8.9 | 1027 | 42.4 | 1025 | 45.7 |
| Part Time | 490 | 42.5 | 331 | 13.7 | 302 | 13.5 |
| **Unemployed (all that apply)** | | | | | | |
| Student | 556 | 48.3 | 71 | 2.9 | 77 | 3.4 |
| Retired | 1 | .1 | 514 | 21.2 | 387 | 17.3 |
| Fulltime Caretaker | 5 | .4 | 112 | 4.6 | 40 | 1.8 |
| Other (Homemaker)* | 0 | 0 | 6 | .2 | 111 | 4.9 |
| Other (Disabled)* | 1 | .1 | 104 | 4.3 | 153 | 6.8 |
| Other (Self-employed)* | 0 | 0 | 6 | .2 | 22 | .9 |
| Other | 13 | 1.1 | 124 | 5.1 | 3 | .1 |

**Notes.** 2022 Student *N* = 1152. 2022 SCRI *N* = 2420. 2023 SCRI *N* = 2243.

*Denotes a coded category added based on open-text "other" responses within that variable in 2022, participants in these added categories are still included in "other" for 2022. Homemaker, Disabled, and Self-employed were added as answer options in 2023 (Employment). Percentage reported is out of the total *N* for each sample.

closer than casual friends. Across each sample, a small, but notable, percentage of friends were kin: romantic partners were the most common (4.7% to 6.4%), but siblings (2.8% to 4.4%) were also common. As Table 4 shows, family was slightly more likely to be included in the SCRI sample compared to the student sample, particularly in the case of parent and child. Importantly, because categories were not mutually exclusive, individuals could list someone as both family *and* friend.

The length of friendship varied widely, although SCRI participants' most common length of friendship was 20+ years and students' most common length was 5–10 years. Friends lived together infrequently, and living with friends was more common among college students. Friends often lived in the same town/city (30.7% to 42.2%). Long distance friendships were still quite common (> 40% lived in a different town/city or further away). Friends met in a variety of places. The most common meeting place was school, particularly for students (65.3%). Participants from the SCRI samples were more likely than student participants to report meeting friends through work (12.5% to 20.2%) and through other friends (12.6% to 17.1%). Meeting friends online was somewhat rare (~5.8%) and occurred at a similar rate across all samples.

**Table 8.** AFP 2022 and 2023 study friend demographics Part I.

| Variable | 2022 Student Sample | | 2022 SCRI Sample | | 2023 SCRI Sample | |
|---|---|---|---|---|---|---|
| **Friend's Age** | N | % | N | % | N | % |
| 18 years old | 1747 | 26.2 | 371 | 4.2 | 400 | 3.9 |
| 19–24 years old | 4447 | 66.7 | 980 | 11.0 | 1241 | 12.1 |
| 25–30 years old | 263 | 3.9 | 1016 | 11.4 | 1480 | 14.4 |
| 31–40 years old | 98 | 1.5 | 1810 | 20.3 | 2270 | 22.1 |
| 41–50 years old | 59 | .9 | 1341 | 15.0 | 1645 | 16.0 |
| 51–60 years old | 39 | .6 | 1405 | 15.7 | 1414 | 13.7 |
| 61+ years old | 14 | .2 | 2013 | 22.4 | 1838 | 17.9 |
| **Friend's Gender** | | | | | | |
| Female | 3703 | 55.5 | 4875 | 54.6 | 5672 | 55.1 |
| Male | 2816 | 42.2 | 3889 | 43.5 | 4426 | 43.0 |
| Nonbinary* | 46 | .7 | 7 | .1 | 77 | .7 |
| Transgender Male | 35 | .5 | 64 | .7 | 48 | .5 |
| Transgender Female | 16 | .2 | 52 | .6 | 46 | .4 |
| Don't Know | 25 | .4 | 30 | .3 | 18 | .2 |
| Prefers to be identified as... | 73 | 1.1 | 26 | .3 | 1 | .0 |
| **Friend's Sexual Orientation** | | | | | | |
| Heterosexual | 5352 | 80.3 | 7948 | 88.9 | 8882 | 86.3 |
| Bisexual | 606 | 9.1 | 411 | 4.6 | 531 | 5.2 |
| Gay | 137 | 2.1 | 221 | 2.5 | 286 | 2.8 |
| Lesbian | 107 | 1.6 | 175 | 2.0 | 207 | 2.0 |
| Don't Know | 353 | 5.3 | 144 | 1.6 | 260 | 2.5 |
| Pansexual* | 40 | .6 | 8 | .1 | 66 | .6 |
| Asexual* | 20 | .3 | 3 | .1 | 45 | .4 |
| Prefers to be identified as... | 113 | 1.7 | 37 | .4 | 11 | .1 |
| **Friend's Ethnicity** | | | | | | |
| Hispanic/Latino/a/x | 1051 | 15.8 | 1049 | 11.7 | 1245 | 12.1 |
| Not Hispanic/Latino/a/x | 5616 | 83.9 | 7887 | 88.3 | 8372 | 81.4 |
| Don't know | – | – | – | – | 671 | 6.5 |
| **Friend's Race (all that apply)** | | | | | | |
| White/Caucasian | 4452 | 66.5 | 6411 | 71.7 | 7640 | 74.3 |
| Black/African American | 603 | 9.0 | 1471 | 16.5 | 1495 | 14.5 |
| Asian | 1165 | 17.4 | 457 | 5.1 | 421 | 4.1 |
| Other (Hispanic)* | 339 | 5.1 | 347 | 3.9 | 177 | 1.7 |
| American Indian/Alaskan Native | 87 | 1.3 | 274 | 3.1 | 192 | 1.9 |
| Native Hawaiian/Pacific Islander | 197 | 2.9 | 124 | 1.4 | 112 | 1.1 |
| Don't know | | | | | 419 | 4.1 |
| Prefers to be identified as... | 552 | 8.2 | 438 | 4.9 | 240 | 2.3 |

**Notes.** 2022 Student Friend *N* = 6667. 2022 SCRI Friend *N* = 8936. 2023 SCRI Friend *N* = 10288.

*Denotes a coded category added based on open-text "other" responses within that variable in 2022, participants in these added categories are still included in "other" in 2022. These responses were added as formal options in 2023: Nonbinary (gender), Pansexual and Asexual (Sexual Orientation). "Don't know" was not an option for race/ethnicity in 2022.

Finally, Table 9 reports on participants' overall network composition and their experiences making new friends and losing friends. The most notable change between 2022 and 2023 was many more participants reporting having made a new friend in 2023 (33.9% in 2022 vs. 51.9%

**Table 9. AFP 2022 and 2023 study friendship network characteristics.**

| Variable | 2022 Student Sample | | 2022 SCRI Sample | | 2023 SCRI Sample | |
|---|---|---|---|---|---|---|
| | N | % | N | % | N | % |
| **How well do your friends know each other**[*] | | | | | | |
| All my friends know each other | 426 | 37.4 | 790 | 33.7 | 490 | 22.2 |
| Some of my friends know each other | 586 | 51.5 | 895 | 38.2 | 1221 | 55.2 |
| Few of my friends know each other | 95 | 8.3 | 371 | 15.8 | 362 | 16.3 |
| None of my friends know each other | 17 | 1.5 | 160 | 6.8 | 106 | 4.8 |
| Does not apply, I only have 1 friend | 14 | 1.2 | 127 | 5.4 | 31 | 1.4 |
| **When did you last make a new friend?** | | | | | | |
| Last 12 months | 990 | 87.0 | 695 | 33.9 | 1165 | 51.9 |
| Last 2 years | 95 | 8.3 | 306 | 14.9 | 354 | 15.8 |
| Last 3 years | 25 | 2.2 | 174 | 8.5 | 153 | 6.8 |
| Last 4 years | 10 | .9 | 136 | 6.6 | 87 | 3.9 |
| Last 5 years | 6 | .5 | 175 | 8.5 | 126 | 5.6 |
| Last 6 years | 2 | .2 | 59 | 2.9 | 31 | 1.4 |
| Last 7 years | 2 | .2 | 42 | 1.7 | 30 | 1.3 |
| Last 8 years | 0 | 0 | 38 | 1.9 | 17 | .8 |
| Longer than 8 years ago | 5 | .4 | 370 | 18.1 | 238 | 10.6 |
| Never | 3 | .3 | 53 | 2.6 | 42 | 1.9 |
| **Have you lost touch with a friend in the past year?**[**] | | | | | | |
| Yes | 837 | 73.6 | 765 | 37.4 | 804 | 35.9 |
| No | 301 | 26.4 | 1280 | 62.6 | 1434 | 64.1 |
| **(*if yes*): Was it more than one friend?**[***] | | | | | | |
| Only one friend | 246 | 29.4 | 332 | 43.4 | – | – |
| More than one friend | 591 | 70.6 | 433 | 56.6 | – | – |

**Notes.** 2022 Student $N$ = 1138. 2022 SCRI $N$ = 2048. 2023 SCRI $N$ = 2243. [*]Only asked for participants who had 1 or more friends. [**]In 2023 this question was re-worded to ask: In the past year, have you had a friendship end? [***]This question was not asked in 2023.

in 2023). Student participants were both more likely to report making a new friend and losing touch with a friend in the past year compared to the SCRI participants.

## RQ2: Communication with friends

RQ2 asked about the communicative functions of friendship. Table 10 reports on the frequency of communicating across a variety of media. Across all three samples, participants were most likely to talk to their friends in-person (63.8% to 66.4% at least once a month). Voice calls (58.6% to 67% at least once a month) and texting (53.3% to 74% at least once a month) were also popular. These three ways of communicating were the most common ways to communicate for all participants. Video calls (60.4% at least once a month) were much more likely to be used among the student sample, and email was more common among the SCRI samples (23.5% to 43.1% at least once a month). Contact through social media was highest for the 2023 SCRI sample (60.8% at least once a month). In the case of video calls, email, and social media, however, frequency of contact may have been due to a small number of participants relying on each channel very frequently, because the modal response was zero.

## RQ3 & RQ4: Well-being and friendship quality

RQ3 asked about how Americans feel about the quality of their friendships, and RQ4 explored their general social health and well-being. The AFP used seven measures of social health and

**Table 10. AFP 2022 and 2023 study communication frequency with friends.**

| Variables | 2022 Student Sample | | | 2022 SCRI Sample | | | 2023 SCRI Sample | | |
|---|---|---|---|---|---|---|---|---|---|
| | M(SD) | Mode (%) | At Least Once a Month (%) | M(SD) | Mode (%) | At Least Once a Month (%) | M(SD) | Mode (%) | At Least Once a Month (%) |
| In-Person | 3.63 (2.21) | 3(22.7) | 66.4 | 3.70 (2.47) | 3(15.3) | 63.8 | 3.90 (2.44) | 2(16.6) | 64.8 |
| Voice Call | 3.65 (2.43) | 3(17.5) | 67.0 | 3.49 (2.46) | 3(14.5)* | 61.3 | 3.41 (2.53) | 1(20.9) | 58.6 |
| Video Call | 3.37 (2.65) | 0(26.2) | 60.4 | 2.14 (2.59) | 0(48) | 37.1 | 2.05 (2.60) | 0(52.1) | 35.0 |
| Text Messaging | 2.92 (1.84) | 1(26.3) | 53.3 | 3.08 (2.32) | 3(16.7) | 55.3 | 4.37 (2.47) | 5(13.3) | 74.0 |
| Social Media | 2.65 (1.94) | 1(25.7) | 46.4 | 2.40 (2.34) | 0(31.3) | 42.5 | 3.65 (2.87) | 0(28.7) | 60.8 |
| Email | 1.11 (2.47) | 0(80.7) | 17.1 | 2.46 (2.73) | 0(45.3) | 43.1 | 2.40 (1.75) | 0(56.0) | 23.5 |

**Notes.** 2022 Student Friend $N$ = 6667. 2022 SCRI Friend $N$ = 8936. 2023 SCRI Friend $N$ = 10288. Single item measures asking to think about their communication through each modality in the past year, reporting options were: 0 = never, 1 = once in the last year, 2 = a few times in the last year, 3 = once a month, 4 = every other week, 5 = weekly, 6 = few times a week, 7 = daily, and 8 = several times a day. *Voice Calls were bi-modal for 2022 SCRI sample, with 14.5% reporting never and 14.5% selecting once a month, however two additional participants selected once a month.

global well-being drawn from the existing literature: life satisfaction, loneliness, state connection, state disconnection, perceived stress, companionship, and perceived social support. The measure source, rating scale, reliability, means, and standard deviations are provided in Table 5. There was a great deal of similarity between the two SCRI samples, with perceived social support slightly higher for 2022 and stress and state disconnection being somewhat higher in 2023. Students reported more connection and companionship and more disconnection and loneliness compared to the SCRI samples.

Table 3 reports means, standard deviations, and percent agreement for questions related to friendship experiences and how participants felt in a broad sense. In 2022, there were three questions about friendship during the COVID-19 pandemic. These questions were created by the co-authors based on prior research conducted during the pandemic. In 2023 the item "I am uncertain how to interact around my friends" was added. For items that were asked across both years, answers were largely stable, including satisfaction with the number of friends they have (75.4 to 76.4%) and satisfaction with time spent with friends (51.1 to 58.8%).

## Conclusions

The first two years of the AFP survey offer a rich and detailed picture of the state of American friendships and their social health. This study revealed several notable trends that have implications for understanding the number of friends, the composition of friendships, social life, and lifespan influences on friendship within the United States. The results also speak to the importance of the methodological choices researchers make when studying friendship and social interaction.

To begin, the present study suggests that Americans have on average four or five friends, which is very similar to the number of friends in estimates from 1970 to 2015 [16, 43]. However, that estimate varied between 2022 and 2023. In the name generation task, participants could identify a maximum of seven friends in 2022 and six friends in 2023, and some participants listed the maximum number of names. In 2023, we added a single item question asking respondents to name the number of friends they had (including a category of 51+ friends),

and 51.2% of participants indicated they had six or more friends. These results suggest that four to five friends may well be an undercount of Americans' total number of friends. In other words, had respondents been given the option of listing more friends than a limit of seven, they would likely do so [20]. However, there is reason to believe that the laboriousness of the name generation task may have reduced some participants' willingness to identify six or seven friends in 2022 [19]. In this regard, having the number of friends identified asked in two different ways—with first an overall count followed by the name generation task—can help to quantify both the broader social network and closer ties, striking a balance between the two.

The number of Americans reporting having no friends was < 3% for the entire sample, which is consistent with past estimates for adolescents (~1%) [44] and adults (1.7%) [19, 43]. Over 40 years, across a broad range of surveys and methods for counting, less than 5% of Americans report having no friends [16]. The results of the AFP survey suggest recent concerns shared in the public sphere about rising rates of friendlessness [14, 15] may be overstated and could be an artifact of data collection methods, data management, participant burden, and/or response options [18–20].

Past researchers [18–20] have also recommended that if the goal is to identify a comprehensive social network, name listing tasks need to approach the question in several ways. In the present study, asking people who said they had no friends to list the names of those with whom they have "pleasant and enjoyable interactions" decreased the "no friend" group by 1.2% to 3.9% across the three samples (2022 student and 2022 and 2023 SCRI samples). Once participants listed names, two-thirds of the "pleasant and enjoyable interactions" group ended up identifying at least one friend, despite having said they had no friends in an earlier part of the survey. Furthermore, in 2023, the addition of the "new friend" question increased the new names listed for 28.5% of respondents, increasing the total friend count, confirming the value of repeating name generation tasks with specific prompts.

The methods used for the AFP may have also uncovered a valuable anchoring question when counting the number of friends. As previously noted, the average number of friends reported in the name generation task increased from 2022 to 2023. In 2022, only 33.7% listed five or more friends but in 2023, 72.0% of participants listed five friends. When responses to the 'new friend' prompt were added, many added a sixth. We believe this may stem from a methodological choice to remove the open-text question asking participants to reflect on "what is a friend" in 2022 and the addition of an ordinal question asking about the total number of friends in 2023 prior to name listing. When participants indicated the total number of their friends, which could have been as high as 51+ friends, they may have felt obligated to be consistent in their responses. For example, participants who indicated they had five or more friends in response to the ordinal question would need to list five or six names to be internally consistent in their answers. By contrast, reflecting on what it means to be a friend may have narrowed people's perception of who to "count" as a friend in 2022, leading to fewer names provided in the subsequent name generation task. In other words, the shift to the ordinal question in 2023 may have provided the opportunity to integrate a more flexible or less narrow definition of friendship. Another indication that this choice of initial survey question mattered can be seen in the increased number of casual connections reported in 2023, when participants were first asked to reflect on their total number of friends (Table 8), as well as the greater number of participants who indicated they had made a new friend in 2023 (51.9%) as compared to 2022 (33.9%).

Beyond the methodological question of how to determine the true number of friends someone has, the initial AFP data suggests that who people count as a friend often includes other relationships in their life, including family and romantic partners. Indeed, friends are often siblings, parents, children, and more distant relatives (i.e., family were 9%-10% of all friends

listed). For adults, romantic partners were often listed as friends (5.6%). Studies that forbid identifying family or spouses as friends may depress the number of friends counted and, more importantly, may miss an important companionate relationship that goes beyond kinship. The AFP purposefully kept the nature of what is a friend broad, allowing for a more representative measure of sociality, which is a major contributor of well-being [7].

Past research sometimes narrowly defines friends as only "close" or "best" friends [14, 15]. However, there is increasing evidence that interactions with weak ties (e.g., casual friends) can be supportive of health and wellness [45,46]. In the present investigation, 55%-60% of names listed were best or close friends, but this leaves a considerable number of "just" friends (~23%). Friendship closeness is influenced by various factors, including participants' gender [47] and geographic access, which suggests researchers might use caution before not counting (or discounting) casual friends.

Another point of interest is that although some Americans lived with their friends (including family who they also consider friends), this was uncommon for all groups except students. However, many people live geographically near their friends: nearly half live in the same town or neighborhood (47.7%). Geographically nearby friends are as common as friends who live in a different city, state, or country as their friends; that is, long distance friendship is quite common. Participants favored voice calls and FtF conversation to keep in touch with friends nearby and far away: 64% spoke at least monthly for FtF and 60% made voice calls at least monthly. Text messaging was also a popular form of keeping in touch. However, email, video calls, and social media varied between years and groups. Given that the modal response for using those media was "never", their usage appears to be influenced by a small set of heavy users rather than being common among all participants.

Over a third of individuals in the SCRI samples had lost touch with a friend in the last year (35.9% to 37.4%). It was twice as common for students to report having lost touch with a friend (73.6%). Students were also much more likely to lose touch with more than one friend, compared to the SCRI sample. Given that college students are making a major life transition that includes forming a new social group, possibly in a new town or state, losing touch with old friends is understandably common. This also confirms prior research that there is more turnover in individuals' social networks than we may realize [48].

Finally, survey responses suggested that overall, Americans were generally on good terms with their friends and satisfied with their friendships, though in many cases there was evidence of wanting to have more time or better-quality friendships, too. Similar to Goddard [14], over 75% were satisfied with the number of friends they had and about two-thirds were satisfied with how they had been maintaining their friendships. A very high portion of respondents (84% to 95.4%) felt they had friends who celebrate their good news and very few respondents reported not getting along with their friends (only 3.7% to 9.9%). This paints a positive view of the health of American friendship. However, there was also a clear sense that Americans longed for greater closeness with their friends. Over 40% felt they were not as close to their friends as they would like, and less than half felt that they were satisfied with the amount of time they had with friends. Surprisingly, this was true for college students and adults alike. Roughly the same portion of the SCRI samples and college students (58.3% vs. 65.6%) felt that it was easier to make friends at another time of life. Plausibly both are referring to high school age and younger, or perhaps a longing for an easier time making friends is a universal sentiment. Indeed, nearly half agreed that it is difficult to make friends, with college students (56.2%) agreeing more frequently than the SCRI samples (46.6% to 48.9%). It is possible this perception is due to the lingering effects of the COVID-19 pandemic. Indeed, some SCRI participants in 2022 felt the pandemic led to lost friends (24.9%) or missing out on making new friends (36.4%), with college students much more likely to feel that way (42.5% and 57.7%

respectively). On the flip side, friendships were also strengthened during the pandemic (41.2% for SCRI sample and 51.6% for college students; Table 3).

## Limitations & future directions

Although SCRI provided access to a diverse and dispersed sample and although there is evidence that Lucid sampling matches the demographics of Americans [25], it cannot be said to be representative of all Americans. College students are also not representative of all Americans between 18–25. Consequently, it is possible that the findings attributed to young people may be due to their education level or income. How this may have influenced the findings of the present investigation is unclear, as the role of education and income on social well-being is mixed and sometimes counter intuitive (e.g., income is positively associated with loneliness [49]; education is positively associated with social network size [19]). All information about participants' friends were gathered from the participants themselves and may also not be accurately reported. Additionally, the reciprocity of each friendship is unknown. Finally, the AFP allowed for the inclusion of individuals who may not be traditionally included in the definition of friend (e.g., family), which can contribute to conceptual fuzziness; however, this was a purposeful choice as it allowed for a more representative measure of social interaction and social networks.

Future research is also needed to assess whether our suppositions about why more friends were listed in 2023 vs. 2022 has merit. This is not simply an empirical question, as it highlights the importance for researchers in considering what a friend is, and whose definition of friendship guides the research. We maintain that it is individuals' own interpretations of friendship that matter; however, it seems that priming them to think about what makes a friend has implications for how they respond to survey questions about their friendship networks.

We will continue to collect data from representative samples to track the status of American friendship over time. Additionally, in fall 2023, the panel study component of the AFP was initiated. Following the college sample collected in 2022, the panel design will be able to examine whether changing conditions of friendship predict changes in well-being over time. Although there is clear evidence that well-being and friendship are related, a longitudinal approach will determine if changes in friendship in the prior year can account for changes in loneliness or connection in the future or vice versa. That is, how does the gain and loss of friends affect well-being? For example, there is evidence that adolescents who can strengthen one friendship from year to year can mitigate future social withdrawal [44]. The panel component of the AFP will explore when and why friendship is a protective barrier to loneliness and bolstering companionship, offering empirical evidence about the steps individuals can take to address rising rates of loneliness.

## Supporting information

**S1 File. Additional demographic information.**
(DOCX)

## Author Contributions

**Conceptualization:** Natalie Pennington, Jeffrey A. Hall, Amanda J. Holmstrom.

**Data curation:** Natalie Pennington, Jeffrey A. Hall, Amanda J. Holmstrom.

**Formal analysis:** Natalie Pennington, Jeffrey A. Hall, Amanda J. Holmstrom.

**Funding acquisition:** Natalie Pennington, Jeffrey A. Hall, Amanda J. Holmstrom.

**Investigation:** Natalie Pennington, Jeffrey A. Hall, Amanda J. Holmstrom.

**Methodology:** Natalie Pennington, Jeffrey A. Hall, Amanda J. Holmstrom.

**Project administration:** Natalie Pennington, Jeffrey A. Hall, Amanda J. Holmstrom.

**Writing – original draft:** Natalie Pennington, Jeffrey A. Hall, Amanda J. Holmstrom.

**Writing – review & editing:** Natalie Pennington, Jeffrey A. Hall, Amanda J. Holmstrom.

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
