## [Decision Letter · Decision Letter 0]

5 Apr 2024

PONE-D-24-01101The American Friendship Project: A Report on the Status and Health of Friendship in AmericaPLOS ONE

Dear Dr. Hall,

Thank you for submitting your manuscript to PLOS ONE. After careful consideration, we feel that it has merit but does not fully meet PLOS ONE’s publication criteria as it currently stands. Therefore, we invite you to submit a revised version of the manuscript that addresses the points raised during the review process.

 In order to be be published, the manuscript requires a major revision. The current version, as noted by reviewer 1, lacks direction and reads more like a report than a research paper. Also, the minor comments by reviewer 2 need to be addressed.

We look forward to receiving your revised manuscript.

Kind regards,

Srebrenka Letina, Ph.D.

Academic Editor

PLOS ONE

Journal Requirements:

3. Thank you for stating the following financial disclosure: "University of Kansas GRF #2177080 (2022); University of Nevada, Las Vegas Department of Communication Studies (2023); Michigan State University ComArtSci Research and Creative Incubator and Accelerator (CRCIA) award (2022-2025)"

6. Please ensure that you include a title page within your main document. You should list all authors and all affiliations as per our author instructions and clearly indicate the corresponding author. 

7. We notice that your supplementary tables are included in the manuscript file. Please remove them and upload them with the file type 'Supporting Information'. Please ensure that each Supporting Information file has a legend listed in the manuscript after the references list.

Additional Editor Comments:

Major revision is required.

Reviewers' comments:

Reviewer's Responses to Questions

**Comments to the Author**

1. Is the manuscript technically sound, and do the data support the conclusions?

Reviewer #1: Partly

Reviewer #2: Yes

2. Has the statistical analysis been performed appropriately and rigorously? 

Reviewer #1: I Don't Know

Reviewer #2: Yes

3. Have the authors made all data underlying the findings in their manuscript fully available?

Reviewer #1: Yes

Reviewer #2: Yes

4. Is the manuscript presented in an intelligible fashion and written in standard English?

Reviewer #1: Yes

Reviewer #2: Yes

5. Review Comments to the Author

Reviewer #1: The paper covers an important topic, and utilises a novel, comprehensive dataset of friendship and social connection in the United States. Although I really wanted to like the paper, I found myself struggling to decipher the research questions and the main premise of the paper. Instead, I felt like I was reading a project report that was intended to summarise the methods and data, without any concrete research questions or hypotheses.

As a result, my critical concern is around the framework of the introduction. The authors state that they are focused on five aspects of social health, but never articulate why these five were chosen (aside from a reference to Holt-Lunstad’s 2012 work), or what they expected to find. The introduction spent considerable time discussing the need for detailed friendship data (and was compelling), but then never materialised into specific questions that needed to be addressed with the data.

Though all of the survey samples, years, and components were interesting, the structure of the introduction left me confused as to what the paper was actually about (aside from describing the project). This is perhaps due to the author’s goal of ‘introducing the AFP and reporting on the first two years’ findings from baseline years (2022-2023)’, which resulted in a paper that read like a summary of all of the work, rather than a research manuscript. As a reviewer, I am conflicted by the paper – the project is commendable, and I look forward to reading some of the outputs - but I struggled to review this submission due to the lack of clear direction.

Reviewer #2: Thank you for the opportunity to review such a strong and well-informed manuscripts. I have a few minor revisions that I hope you'll consider.

- On Page 3, Paragraph 2 you state the term "discussion partners", and it would be beneficial to include a definition here. How does that differ from a (casual) friend?

- On Page 3, Paragraph 3 you discuss differentiating between a friend, close friend, and best friend. It would be beneficial to also include a short description of strong- and weak-tied dyads. Especially because you discuss weak ties on Page 19, Paragraph 2.

- Strong thesis and purpose of your study (Page 4, Paragraph 1).

- On Page 4, Paragraph 2 it would be beneficial to state which disciplines have (if any) researched friendship scholarship.

- In the first paragraph of your Methods section (Page 5), please double check that your dates are consistent (e.g., July 21, 2022 and September [not abbreviated]).

- On Page 6 in the Demographic paragraph, you state at the end that you yielded 13 options. However, when I counted on Table 4, I found 14?

- On Page 7 Paragraph 1, I found difficulty with keep up with what the participant was prompted with if they didn't have any friends. I think it would be clarifying to not include so many direct quotes from your survey, and instead write it out yourself in a more succinct and prompt manner.

- On Page 7 Paragraph 2 I am failing to see how the open-ended question of "What is a friend?" could then be replaced with "Identify the total number of friends they believed they had." Could you go into depth about why this choice was made?

- On Page 13 Paragraph 2, please make sure all numbers have the percentage sign after (for consistency).

- On Page 14 Paragraph 1 (last sentence), you mention that individuals could list someone who is both family and friend. This completely adds to the "fuzziness" of what a friend is, and I think it would be beneficial to discuss this further in your limitations (and bring the focus back to this section).

- Tables are extremely well done!

Again, thank you for this opportunity. There is some valuable information here, and I'm happy to be part of this publishing opportunity.

6. PLOS authors have the option to publish the peer review history of their article (what does this mean?). If published, this will include your full peer review and any attached files.

Reviewer #1: No

Reviewer #2: **Yes: **Brianna L. Avalos

---

## [Author Response · Author response to Decision Letter 0]

25 Apr 2024

Reviewer 1 Comments to the Author

Reviewer #1: The paper covers an important topic, and utilises a novel, comprehensive dataset of friendship and social connection in the United States. Although I really wanted to like the paper, I found myself struggling to decipher the research questions and the main premise of the paper. Instead, I felt like I was reading a project report that was intended to summarise the methods and data, without any concrete research questions or hypotheses. As a result, my critical concern is around the framework of the introduction. The authors state that they are focused on five aspects of social health, but never articulate why these five were chosen (aside from a reference to Holt-Lunstad’s 2012 work), or what they expected to find. The introduction spent considerable time discussing the need for detailed friendship data (and was compelling), but then never materialised into specific questions that needed to be addressed with the data.

Thank you for your question. We have consulted with the editors and believe that our work qualifies as an research article based on the standards of the journal. However, we see this as an opportunity to strengthen and clarify our intentions in this manuscript. Specifically, we took the following steps: 

We have sought to better clarify the goals and takeaways from the data reported here. We add further justification to focus on these aspects of social health and offer four research questions that are directly tied to them to better frame the study results (introduced page 5): 

 RQ1: What is the structure of Americans’ friendships (e.g, number, characteristics, initiation, longevity, closeness)?

 RQ2: What are the communicative functions of Americans’ friendships (e.g., frequency, modality, proximity)?

 RQ3: How do Americans perceive the quality of their friendship (e.g., support, satisfaction, time, closeness)?

 RQ4: What is the state of Americans social health and well-being (e.g., connection, companionship, loneliness, disconnection)? 

We have also adjusted the text of the results and discussion (pages 14-25) to account for these RQs. We go into such descriptive detail about the data we have in part because we are also making the data publicly available upon publication (OSF), and want to encourage others to reflect on what we have found to explore their own questions about friendship, including building on the work we have started here and will continue to do.

Though all of the survey samples, years, and components were interesting, the structure of the introduction left me confused as to what the paper was actually about (aside from describing the project). This is perhaps due to the author’s goal of ‘introducing the AFP and reporting on the first two years’ findings from baseline years (2022-2023)’, which resulted in a paper that read like a summary of all of the work, rather than a research manuscript. As a reviewer, I am conflicted by the paper – the project is commendable, and I look forward to reading some of the outputs - but I struggled to review this submission due to the lack of clear direction.

We appreciate this question, and have sought to address that by clarifying RQs for the manuscript (see comment above). We consulted with PLOS ONE regarding the submission types considered for publication and where this manuscript fits. We believe this meets the requirements for a research article, as it reports on data collected and descriptive results of that data for analysis. In this revision we have sought to make that clearer through the addition of the RQs shared, which reflect back the facets of social health that we have argued in favor of studying.

Based on the reviewer feedback, restructured our findings around these research questions, which has improved the clarity and quality of the manuscript. We have also sought to strengthen the wording and clarity of the findings throughout. 

Reviewer 2 Comments to the Author (Brianna L. Avalos) 

Thank you for the opportunity to review such a strong and well-informed manuscripts. I have a few minor revisions that I hope you'll consider.

- On Page 3, Paragraph 2 you state the term "discussion partners", and it would be beneficial to include a definition here. How does that differ from a (casual) friend?

We were referring to the “with whom do you discuss important matters” measure that has been used by the General Social Survey since 1985. We have made this explicit in the text (pg 2-3).

- On Page 3, Paragraph 3 you discuss differentiating between a friend, close friend, and best friend. It would be beneficial to also include a short description of strong- and weak-tied dyads. Especially because you discuss weak ties on Page 19, Paragraph 2.

We have added a brief note on strong and weak ties on page 3.

- Strong thesis and purpose of your study (Page 4, Paragraph 1).

Thank you for your feedback.

- On Page 4, Paragraph 2 it would be beneficial to state which disciplines have (if any) researched friendship scholarship.

We have added a statement in this paragraph to acknowledge the different disciplines from which we built out the AFP work (see page 4).

- In the first paragraph of your Methods section (Page 5), please double check that your dates are consistent (e.g., July 21, 2022 and September [not abbreviated]).

This has been corrected to fully spell out September (now page 6).

- On Page 6 in the Demographic paragraph, you state at the end that you yielded 13 options. However, when I counted on Table 4, I found 14?

We apologize for this confusion. The final option we believe you are counting in each case is that we had “none of the above” for those who clicked none of the other options. So in 2022 there were 12 possible changes or none (13 lines) and in 2023 there were 13 possible changes or none (14 lines) (see Table 4, page 17). We have also clarified this in text (see page 8, formerly page 6).

- On Page 7 Paragraph 1, I found difficulty with keep up with what the participant was prompted with if they didn't have any friends. I think it would be clarifying to not include so many direct quotes from your survey, and instead write it out yourself in a more succinct and prompt manner.

We have adjusted this text (see pages 8-9, formerly page 7) to remove the quotes for clarity.

- On Page 7 Paragraph 2 I am failing to see how the open-ended question of "What is a friend?" could then be replaced with "Identify the total number of friends they believed they had." Could you go into depth about why this choice was made?

We agree as presented this may come across as unclear, and have added a brief bit of text on page 9 (formerly page 7) to clarify. The decision to shift these questions between the first year and second year was not that we felt they were equivalent questions, but instead, filled a gap in the 2022 study, which limited our understanding of the volume of friends participants had to only up to 7 names. Changing the question in year two presented us with additional data and provided a point of reference to reflect on how the framing of friendship from the start influenced how people conceptualized friendship (see conclusions, page 25-27). 

- On Page 13 Paragraph 2, please make sure all numbers have the percentage sign after (for consistency).

Thank you for catching this, we have corrected the manuscript to include % in all cases where it should be (see page 18, formerly page 13).

- On Page 14 Paragraph 1 (last sentence), you mention that individuals could list someone who is both family and friend. This completely adds to the "fuzziness" of what a friend is, and I think it would be beneficial to discuss this further in your limitations (and bring the focus back to this section).

While we understand this may contribute to fuzziness, we do currently address this decision in the conclusions of the manuscript. We argue in favor of including family and romantic partners, in part because studies that forbid identifying family as friends may depress the number of friends and may miss an important relationship that goes beyond kinship for some people (conclusions, page 27). We have also added a further note about this in the limitations (see page 30-31).

- Tables are extremely well done!

Thank you!

Again, thank you for this opportunity. There is some valuable information here, and I'm happy to be part of this publishing opportunity.

Thank you for your thoughtful feedback; we believe it has helped to strengthen the manuscript.

---

## [Decision Letter · Decision Letter 1]

6 Jun 2024

The American Friendship Project: A Report on the Status and Health of Friendship in America

PONE-D-24-01101R1

Dear Dr. Hall,

We’re pleased to inform you that your manuscript has been judged scientifically suitable for publication and will be formally accepted for publication once it meets all outstanding technical requirements.

Kind regards,

Srebrenka Letina, Ph.D.

Academic Editor

PLOS ONE

Additional Editor Comments (optional):

Reviewers' comments:

Reviewer's Responses to Questions

**Comments to the Author**

1. If the authors have adequately addressed your comments raised in a previous round of review and you feel that this manuscript is now acceptable for publication, you may indicate that here to bypass the “Comments to the Author” section, enter your conflict of interest statement in the “Confidential to Editor” section, and submit your "Accept" recommendation.

Reviewer #1: All comments have been addressed

Reviewer #2: All comments have been addressed

2. Is the manuscript technically sound, and do the data support the conclusions?

Reviewer #1: Yes

Reviewer #2: Yes

3. Has the statistical analysis been performed appropriately and rigorously? 

Reviewer #1: Yes

Reviewer #2: Yes

4. Have the authors made all data underlying the findings in their manuscript fully available?

Reviewer #1: Yes

Reviewer #2: Yes

5. Is the manuscript presented in an intelligible fashion and written in standard English?

Reviewer #1: Yes

Reviewer #2: Yes

6. Review Comments to the Author

Reviewer #1: Excellent revision. The added research questions and structure made the article more easily understood. Really enjoyable work.

Reviewer #2: Thank you for addressing all of my previous concerns - this is a sound and strong paper. It contains important information that is addressed appropriately.

7. PLOS authors have the option to publish the peer review history of their article (what does this mean?). If published, this will include your full peer review and any attached files.

Reviewer #1: No

Reviewer #2: No
